# A CECT-Based Radiomics Nomogram Predicts the Overall Survival of Patients with Hepatocellular Carcinoma After Surgical Resection

**DOI:** 10.3390/biomedicines13051237

**Published:** 2025-05-19

**Authors:** Peng Zhang, Yue Shi, Maoting Zhou, Qi Mao, Yunyun Tao, Lin Yang, Xiaoming Zhang

**Affiliations:** Medical Imaging Key Laboratory of Sichuan Province, Department of Radiology, Interventional Medical Center, Science and Technology Innovation Center, The Affiliated Hospital of North Sichuan Medical College, Nanchong 637000, China

**Keywords:** computed tomography (CT), radiomics, hepatocellular carcinoma (HCC), overall survival (OS), nomogram

## Abstract

**Objective**: The primary objective of this study was to develop and validate a predictive nomogram that integrates radiomic features derived from contrast-enhanced computed tomography (CECT) images with clinical variables to predict overall survival (OS) in patients with hepatocellular carcinoma (HCC) after surgical resection. **Methods**: This retrospective study analyzed the preoperative enhanced CT images and clinical data of 202 patients with HCC who underwent surgical resection at the Affiliated Hospital of North Sichuan Medical College (Institution 1) from June 2017 to June 2021 and at Nanchong Central Hospital (Institution 2) from June 2020 to June 2022. Among these patients, 162 patients from Institution 1 were randomly divided into a training cohort (112 patients) and an internal validation cohort (50 patients) at a 7:3 ratio, whereas 40 patients from Institution 2 were assigned as an independent external validation cohort. Univariate and multivariate Cox proportional hazards regression analyses were performed to identify clinical risk factors associated with OS after HCC resection. Using 3D-Slicer software, tumor lesions were manually delineated slice by slice on preoperative non-contrast-enhanced (NCE) CT, arterial phase (AP), and portal venous phase (PVP) images to generate volumetric regions of interest (VOIs). Radiomic features were subsequently extracted from these VOIs. LASSO Cox regression analysis was employed for dimensionality reduction and feature selection, culminating in the construction of a radiomic signature (Radscore). Cox proportional hazards regression models, including a clinical model, a radiomic model, and a radiomic–clinical model, were subsequently developed for OS prediction. The predictive performance of these models was assessed via the concordance index (C-index) and time–ROC curves. The optimal performance model was further visualized as a nomogram, and its predictive accuracy was evaluated via calibration curves and decision curve analysis (DCA). Finally, the risk factors in the optimal performance model were interpreted via Shapley additive explanations (SHAP). **Results**: Univariate and multivariate Cox regression analyses revealed that BCLC stage, the albumin–bilirubin index (ALBI), and the NLR–PLR score were independent predictors of OS after HCC resection. Among these three models, the radiomic–clinical model exhibited the highest predictive performance, with C-indices of 0.789, 0.726, and 0.764 in the training, internal and external validation cohorts, respectively. Furthermore, the time–ROC curves for the radiomic–clinical model showed 1-year and 3-year AUCs of 0.837 and 0.845 in the training cohort, 0.801 and 0.880 in the internal validation cohort, and 0.773 and 0.840 in the external validation cohort. Calibration curves and DCA demonstrated the model’s excellent calibration and clinical applicability. **Conclusions**: The nomogram combining CECT radiomic features and clinical variables provides an accurate prediction of OS after HCC resection. This model is beneficial for clinicians in developing individualized treatment strategies for patients with HCC.

## 1. Introduction

Hepatocellular carcinoma (HCC) is the sixth most common cancer and the third leading cause of cancer-related death worldwide. It is a major social and public health challenge in the 21st century [1,2,3]. Surgical resection is one of the most effective means for achieving long-term survival in HCC patients. In recent years, owing to the progress of various systemic and local treatments, the possibility of surgical resection for patients with middle- and advanced-stage HCC has increased [4,5]. The overall survival (OS) of HCC patients after resection is the core indicator for evaluating the disease prognosis and effectiveness of treatment in patients [6]. Among HCC patients, the 5-year postsurgical rate of recurrence, which affects OS, is approximately 70% [7]. Thus, for clinicians, accurate preoperative prediction of disease prognosis after HCC resection is helpful for developing individualized treatment plans to further improve HCC prognosis.

In recent years, studies have shown that clinical variables such as the neutrophil-to-lymphocyte ratio (NLR) and platelet-to-lymphocyte ratio (PLR) are closely related to tumor prognosis [8,9,10]. The radiomic model has important value in the prediction of HCC prognosis in patients [11,12,13], but its “black box” character limits its clinical application [14]. The Shapley additive explanation (SHAP) value is a game theory-based interpretation method that is used to quantify the contribution of each feature to model predictions, thereby improving the interpretability of the model and helping clinicians better understand the model [15,16]. However, to date, few studies have focused on the construction of a prognostic model after HCC resection on the basis of preoperative radiomic features combined with clinical inflammatory factors, and SHAP analysis of the model has been conducted. The aim of this study was to investigate the value of a nomogram based on preoperative enhanced computed tomography (CT) radiomic features combined with clinical variables in predicting OS after HCC resection. Additionally, SHAP interpretability analysis was employed to further quantify the degree of contribution of each risk factor in the optimal model.

## 2. Materials and Methods

### 2.1. General Information

Data on the CT images and clinical features of patients who underwent HCC resection at Institution 1 between June 2017 and June 2021 or at Institution 2 between June 2020 and June 2022 and who had undergone contrast-enhanced CT examination within 4 weeks before surgery were retrospectively collected. Patients from Institution 1 were randomly divided into training and internal validation cohorts at a ratio of 7:3; patients from Institution 2 composed the independent external validation cohort (Figure 1).

The inclusion criteria were as follows: (1) the patient underwent first-time radical resection of liver cancer and had confirmed HCC by postoperative pathology; (2) the patient did not receive any other intervention, such as radiofrequency ablation, interventional therapy, or chemotherapy, before surgery; (3) an enhanced CT scan was performed within 4 weeks before surgery; and (4) the patient was followed up for at least 3 years after surgery.

The exclusion criteria were as follows: (1) the patient was not examined by enhanced CT within 4 weeks before surgery; (2) the CT image quality was poor; (3) the patient had other malignant tumors; or (4) the clinical or follow-up data of the patient were incomplete.

The clinical data of the patients were collected through the hospital electronic records system.

### 2.2. Follow-Up

The survival data of the patients were collected by querying the hospitals’ clinical electronic medical records, outpatient visit records, and telephone follow-up records. The follow-up time of the patients in Institution 1 ended on 30 June 2024, and the follow-up time of patients in Institution 2 ended on 31 December 2024. The endpoint of this study was patient OS, which was defined as the time after HCC resection until the last follow-up or death of patients, and the follow-up time was at least three years.

### 2.3. Screening of Clinical Variables

The clinical data of the patients were analyzed via univariate and multivariate Cox regression to screen out independent predictors for OS after HCC resection.

### 2.4. CT Scan

In Institution 1, plain and contrast-enhanced CT scans were completed for patients via Philips Brilliance (Philips Medical Systems Nederland B.V., Veenpluis 6, 5684 PC Best, The Netherlands), GE Lightspeed VCT (GE Medical SystemsLLC, 3000 North Grandview Blvd., Waukesha, WI 53188, USA), and Siemens Definition AS (Siemens AG, Wittelsbacherplatz 2, DE-80333 Muenchen, Germany) equipment. The scan parameters were set as follows: tube voltage, 120 kV; tube current, 142–250 mA; tube rotation time, 0.50–0.75 s; matrix, 512 × 512; layer thickness, 5 mm; and pitch, 0.8. In Institution 2, patients were scanned preoperatively via GE Lightspeed VCT (GE Medical SystemsLLC, 3000 North Grandview Blvd., Waukesha, WI 53188, USA) or Somatom Definition Flash dual-source CT (Siemens AG, Wittelsbacherplatz 2, DE-80333 Muenchen, Germany). The scan parameters were set as follows: tube voltage, 120 kV; tube current, 210–250 mA; gantry rotation time, 0.50 s; matrix, 512 × 512; slice thickness, 5 mm; and thread pitch, 0.8. After the acquisition of plain scan images, a contrast-enhanced scan was performed. A quantity of 60 to 100 mL of nonionic contrast agent was used at an injection rate of 2.5–3.0 mL/s. The arterial phase and portal venous phase scans were performed 30–35 s and 55–60 s after the injection of contrast agent, respectively.

### 2.5. Radiomics Process

#### 2.5.1. Image Preprocessing, Segmentation, and Radiomic Feature Extraction

All the CT images of all the patients were retrieved in DICOM format from the Picture Archiving and Communication Systems (PACS). Then, the preoperative CT images were imported into 3D-Slicer software in DICOM format for segmentation of the tumor lesions. To minimize the impact of different CT models on the experimental results, the original images were preprocessed before extracting radiomic features. This preprocessing included resampling (voxel size: 1 mm × 1 mm × 1 mm), image gray-level discretization (bin width: 25 HU), and image gray-level normalization to control image noise and standardize the voxel intensity [17,18]. Two radiologists with 3 and 5 years of work experience delineated each lesion layer by layer on plain CT, arterial phase, and portal venous phase images to generate a volume of interest (VOI). Finally, 1130 radiomic features were extracted from the NCE, AP, and PVP images. These included seven classes: tumor morphological features (14 features), first-order statistical features (216 features), a gray-level cooccurrence matrix (GLCM, 288 features), a gray-level run-length matrix (GLRLM, 192 features), a gray-level size-zone matrix (GLSZM, 192 features), a neighborhood gray-tone difference matrix (NGTDM, 60 features), and a gray-level dependence matrix (GLDM, 168 features).

#### 2.5.2. Consistency Assessment

The intraobserver and interobserver consistency of the features were evaluated by determining the intraclass correlation coefficients (ICCs). To calculate the intraobserver ICC, the CT images of 64 patients were randomly selected and segmented twice by radiologist A (PZ) within one month. To calculate the interobserver ICC, the selected CT images were independently segmented by two radiologists (radiologist A and radiologist B (YS)). Features with ICCs < 0.75 were excluded. A feature with an ICC > 0.75 was considered to have good stability and consistency, and this feature was retained and entered the next step of screening.

#### 2.5.3. Feature Screening and Radscore Construction

The obtained radiomic feature data were processed via the Z score statistical method to eliminate the dimensionality differences in the data. Redundant features were removed via the Spearman correlation test. In this step, feature pairs with an absolute Spearman correlation coefficient above 0.8 were removed. The LASSO Cox regression algorithm was used to select the best features for predicting prognosis, and the Radscore was calculated according to the coefficient corresponding to each feature.

### 2.6. Model Development and Evaluation

Cox hazard regression analysis was used to construct a radiomic model, a clinical model, and a radiomic–clinical fusion model. The C-index and time-dependent receiver operating characteristic (time–ROC) curves were used to evaluate the predictive performance of each model. The model with the best performance was visualized as a nomogram, and the calibration curve, decision curve analysis (DCA), and Kaplan–Meier curve were used to evaluate the degree of calibration, clinical practicality, and risk stratification ability of the model (Figure 2).

### 2.7. Statistical Tools and Methods

R (4.4.2) and Python (3.13.1) software were used for the statistical analysis in this study. For clinical quantitative data, normally distributed variables are expressed as means ± standard deviations, whereas nonnormally distributed variables are expressed as medians (interquartile ranges); according to the data distribution, one-way analysis of variance (ANOVA) or the Kruskal–Wallis H test was performed for intergroup comparisons. For categorical variables, intergroup comparisons were performed via the chi-square test or Fisher’s exact test. SHapley Additive exPlanations (SHAP) interpretability analysis of Cox proportional hazards regression models was performed via the Shap package of Python.

## 3. Results

### 3.1. Baseline Information

In total, 202 patients were ultimately enrolled in this study. There were 162 patients from Institution 1, including 141 males and 21 females, and the mean age was 57.4 ± 10.4 years. These patients were randomly divided into a training cohort (112 patients) and an internal validation cohort (50 patients) at a ratio of 7:3. Forty patients from Institution 2, including 31 males and 9 females, were included in an independent external validation cohort; the mean age was 56.45 ± 11.1 years.

Until the cutoff date of follow-up, 64 patients at Institution 1 were alive (39.5%), and 98 patients had died (60.5%); at Institution 2, 17 patients had survived (42.5%), and 23 had died (57.5%). The median follow-up duration for all patients was 59.0 months, and the median survival time was 37.0 months. There was no significant difference in the clinical data of the training cohort, the internal validation cohort, or the external validation cohort (Table 1).

### 3.2. Screening of Clinical Indicators

ROC curves were drawn using the NLR and PLR as survival variables, and the respective cutoff values were determined according to the maximum Youden index. An NLR ≥ 2.77 and a PLR ≥ 141.31 were each assigned as 1 point; otherwise, 0 points were given, and the two scores were added to obtain the NLR-PLR score [19,20]. Univariate Cox proportional hazards regression analysis revealed that the maximum diameter of the tumor, portal vein tumor thrombus, tumor number, AFP, Child–Pugh stage, Barcelona Clinic Liver Cancer (BCLC) stage, NLR, PLR, aspartate aminotransferase (AST), NLR-PLR score, and ALBI grade were significantly associated with OS. Multivariate Cox proportional hazards regression analysis revealed that BCLC stage, NLR-PLR score, and ALBI grade were independent predictors of OS after HCC resection (Table 2).

### 3.3. Radiomic Feature Screening and Radscore Construction

In this study, 1130 features were extracted from the NCE, AP, and PVP images. After consistency analysis, Spearman correlation analysis, and LASSO Cox regression analysis for dimensionality reduction, 14 features with nonzero coefficients were obtained (Table 3). The Radscore was constructed on the basis of the coefficients of each feature. The Mann–Whitney U test revealed that in the training cohort, the internal validation cohort, and the external validation cohort, there were significant differences in the Radscores of surviving patients and nonsurviving patients.

### 3.4. Model Construction and Evaluation

The AUC and C-index values of each model for the prediction of OS at 1 and 3 years after surgery are presented in Table 4. Among the three cohorts, the AUC and C-index of the fusion model were greater than those of the other single models (Table 4, Figure 3).

### 3.5. Evaluation of the Consistency, Clinical Utility, and Risk Stratification of the Combined Model

The fusion model was visualized as a traditional nomogram (Figure 4), and the web-based version of the dynamic nomogram was constructed on the public platform ShinyApps, which can be accessed via https://hcc-resection-ospred.shinyapps.io/HCCDynNomapp/, accessed on 20 March 2025 (Figure 5). Among the three cohorts, the calibration curve revealed that the combined model had a good ability to predict the 1-year and 3-year OS rates after HCC surgery (Figure 6); the DCA curve revealed that the model had a good net gain (Figure 7); the Kaplan–Meier curve [21] revealed that there were significant differences in the OS rates of patients in the high-risk cohort and those in the low-risk cohort; and the OS time of the patients in the high-risk cohort was significantly shorter than that in the low-risk cohort (Figure 8).

### 3.6. SHAP Interpretability Analysis

To further analyze the importance of the features included in the radiomic–clinical Model, SHAP interpretability analysis was used to quantify the contributions of these features. The mean value of each feature in all the samples of the training cohort was calculated as the SHAP value (Figure 9). The results revealed that the Radscore had the greatest influence on the final prediction result in the model (Figure 9A,B), followed by the BCLC stage. Furthermore, we demonstrated that the model can be used to evaluate survival outcomes for two randomly selected samples with different survival outcomes (Figure 9C,D).

## 4. Discussion

Radiomics converts potential pathophysiological information in medical images that cannot be recognized by the human eye into high-dimensional quantitative image features and analyzes the relationships between these features and clinical or genetic data for disease classification and prognosis prediction [22]. Deng PZ et al. constructed a Radscore based on the radiomic features of preoperative contrast-enhanced CT images of 150 HCC patients and combined it with clinical indicators to establish a nomogram to predict postoperative OS after HCC resection. As a result, the C-indices of the training cohort and the validation cohort were 0.736 and 0.774, respectively [21]. Kuang et al. selected 18 radiomic features from the T2WI, AP, PVP, and DP images of 141 preoperative MR images of HCC patients via SelectKBest and LASSO. Combined with the independent clinical risk factors AFP and Ki-67, the multinomial NB classifier was selected to establish a radiomic–clinical pathological factor fusion model to predict the 3-year OS of HCC patients after surgery. The results revealed that the AUCs of the fusion model in the training cohort and the validation cohort were 0.910 and 0.846, respectively [23]. On the basis of MRI and CT radiomics, He Y established a multimodal prediction model to predict disease-free survival (DFS) and OS after HCC resection. The results revealed that the Radscore established by combining CT and MRI data was an independent predictor of DFS and OS after HCC surgery (*p* <0.05); in the CT + MRI + clinical model combined with clinical factors, the C-indices for predicting DFS and OS in the validation cohort reached 0.704 and 0.738, respectively [24]. This study divided the patients from two independent units into a training cohort, an internal validation cohort, and an external validation cohort. Preoperative contrast-enhanced CT images and clinical data were combined to construct a prediction model of OS after HCC surgery. The C-index values of the training, internal validation, and external validation cohorts were 0.789, 0.726, and 0.764, respectively, indicating that the model has good repeatability and generalizability.

As a quantitative scoring tool based on radiomics, the Radscore is currently extensively used to construct various predictive models [17,25,26]. The SHAP analysis results of the fusion model revealed that the Radscore had the highest contribution value. Among the 14 best radiomic features screened in this cohort, 85% were texture features, suggesting that compared with morphological features and first-order features, texture features may be more related to tumor heterogeneity. Previous studies have also shown that a prediction model based on texture features on CT or MR images has an important role in tumor identification, treatment response, and prognosis prediction [27,28,29], which is consistent with the findings of this study.

Additionally, in this study, univariate and multivariate Cox regression analyses revealed that BCLC stage, ALBI grade, and NLR-PLR score were independent predictors of OS in patients after HCC surgery, which was consistent with the findings in the literature [21,30,31,32]. The BCLC staging system is a widely used HCC staging method, and the relevant guidelines provide clear treatment recommendations for each disease stage [33,34].

The prognostic assessment of HCC patients according to the traditional Child–Pugh liver function classification is not ideal, possibly because of its inclusion of subjective evaluation indicators. As an evidence-based scoring system, the ALBI grading system uses only albumin and bilirubin for calculations [35]. Studies have shown that the ALBI grade can improve the prediction of the prognosis of HCC patients [36,37,38,39]. In their large, multicenter study of 2426 HCC patients, Pinato et al. revealed that the ALBI grade was an important predictor of OS after surgical resection, transarterial chemoembolization (TACE), and sorafenib treatment [31].

Studies have shown the role of inflammatory factors such as the NLR and PLR in tumor prognostic prediction models [40,41]. The results of a prognostic study of hepatitis B-associated HCC patients revealed that there was no significant difference in the performance of prediction models based on each single inflammatory indicator, and the performance of a prediction model that integrated multiple inflammatory indicators was significantly better than that of each model that used a single indicator [42]. In this cohort of patients, 65% of the patients had hepatitis B-associated HCC, and univariate Cox proportional hazards analysis revealed that the NLR, PLR, and NLR-PLR scores were significantly associated with OS after HCC surgery. Multivariate analysis revealed that only the NLR–PLR score was an independent risk factor for predicting OS after HCC surgery.

Unlike the findings of the cited studies [43,44], in this study, the maximum tumor diameter or AFP level did not significantly correlate with OS. This may be related to various factors, such as sample size, individual differences in the study population, and treatment options.

Three methods of segmentation are currently used in radiomic analysis to delineate target lesions: manual, semiautomatic, and automatic segmentation. Among them, manual segmentation is the most common method. In this study, manual segmentation was used in accordance with the protocols in a previous study [45,46].

This study has the following limitations. First, although our study was a dual-center study, the sample size was relatively small, which may have affected the reliability of the model. It is necessary to integrate data from multiple sources with larger sample sizes for further validation. Second, because this was a retrospective study spanning a long period, pathological factors such as MVI, degree of tumor differentiation, and gene expression were not included. These data should be included for further verification to improve the results. Third, our follow-up time was set to 3 years, and the 5-year survival rate of patients with tumors is still an important indicator. The follow-up time can be prolonged for further study in the future. Fourth, this study relied only on CT single-modality imaging, and combining MRI or other modalities may improve prediction. Finally, there was no detailed analysis of postoperative data (such as TACE, targeted therapy and immunotherapy), which could affect survival outcomes in this study. These factors need to be included in future studies.

## 5. Conclusions

The prediction model and visualization nomogram based on contrast-enhanced CT radiomic features combined with clinical variables can better predict OS after HCC resection, and this research is expected to provide reference information for clinicians in the development of individualized treatment plans.

## Figures and Tables

**Figure 1 biomedicines-13-01237-f001:**
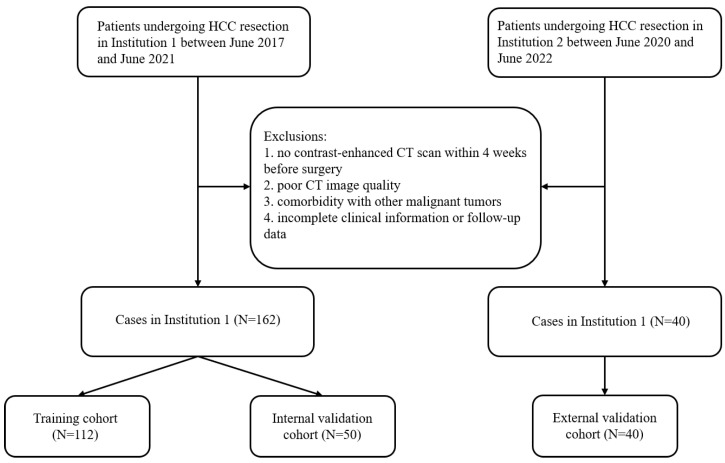
Flowchart of the inclusion and exclusion criteria for the study subjects.

**Figure 2 biomedicines-13-01237-f002:**
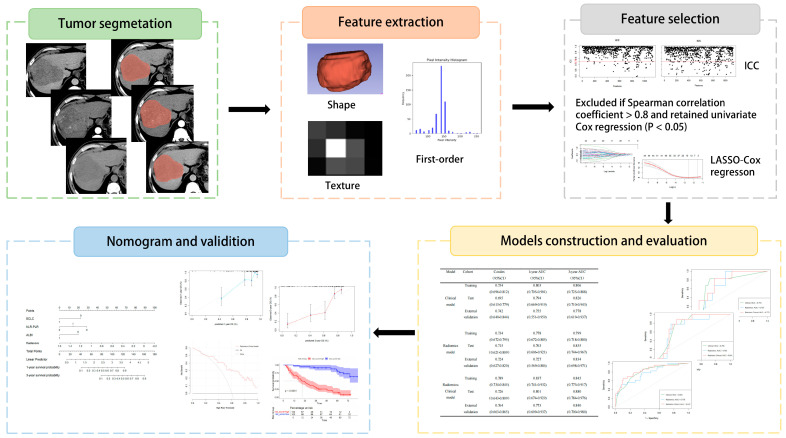
Radiomics workflow diagram. The CT images of all patients were imported into 3D-Slicer software in DICOM format for tumor segmentation. Radiomic features were subsequently extracted from the NCE, AP, and PVP images. Following dimensionality reduction via consistency analysis, Spearman correlation analysis, and LASSO Cox regression, optimal features with nonzero coefficients were retained. Cox hazard regression analysis was used to construct models, and the model with the best performance was visualized as a nomogram.

**Figure 3 biomedicines-13-01237-f003:**
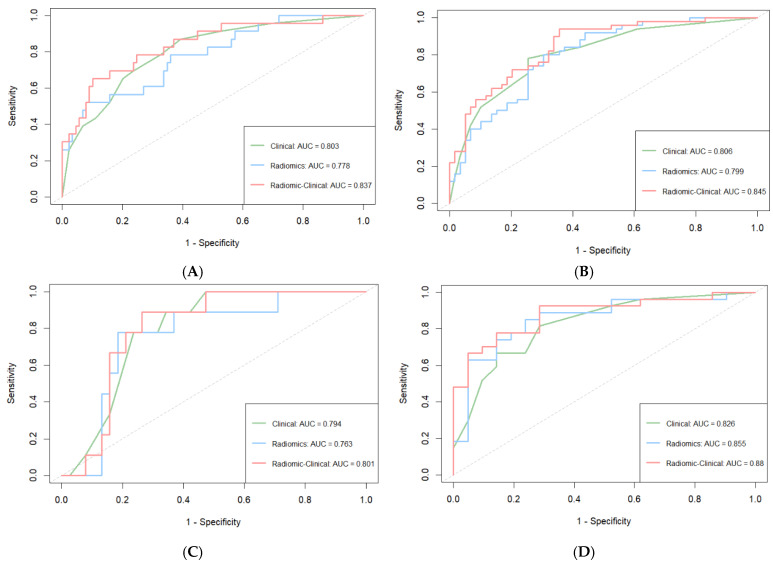
Time–ROC curves for predicting 1-year OS (**A**) and 3-year OS (**B**) in the training cohort, 1-year OS (**C**) and 3-year OS (**D**) in the internal validation cohort, and 1-year OS (**E**) and 3-year OS (**F**) in the external validation cohort.

**Figure 4 biomedicines-13-01237-f004:**
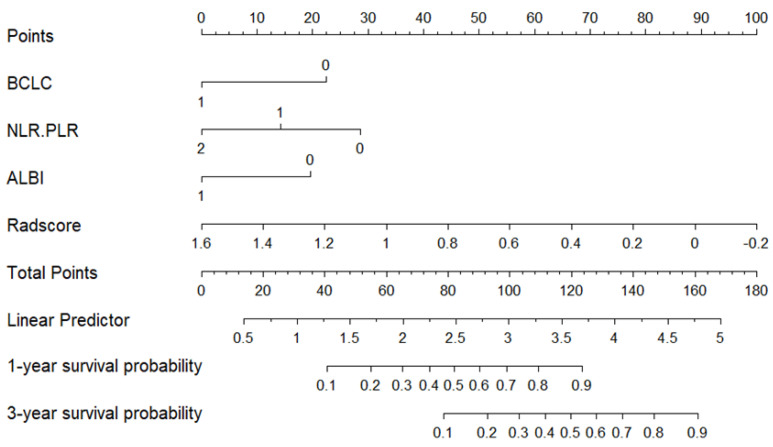
Nomogram of the 1-year and 3-year survival probabilities of patients after HCC surgery according to the radiomic–clinical model.

**Figure 5 biomedicines-13-01237-f005:**
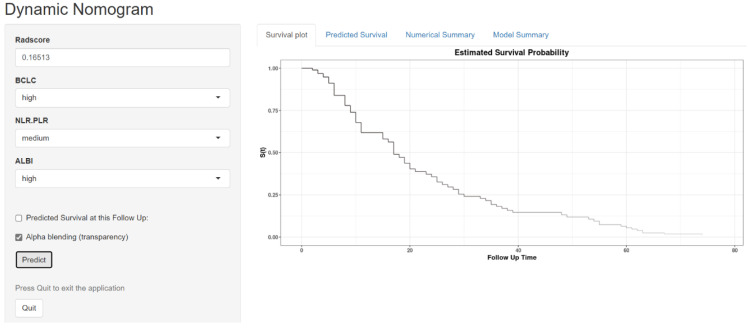
Radiomic–clinical model dynamic nomogram web interface.

**Figure 6 biomedicines-13-01237-f006:**
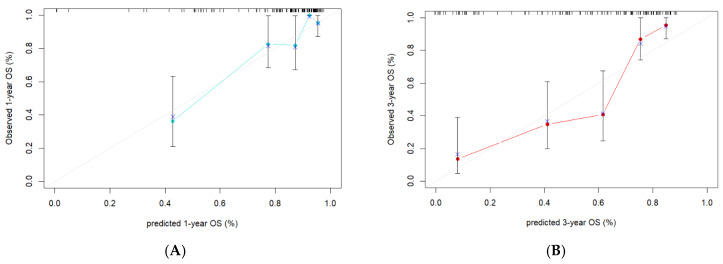
Calibration curves for the ability of the radiomic–clinical model to predict 1-year OS (**A**) and 3-year OS (**B**) in the training cohort, 1-year OS (**C**) and 3-year OS (**D**) in the internal validation cohort, and (**E**) 1-year OS and (**F**) 3-year OS in the external validation cohort. *x*-axis: the predicted 1- and 3-year survival. *y*-axis: the observed 1- and 3-year survival, as estimated by Kaplan-Meier method. Gray lines: the optimal alignment between observed and predicted 1- and 3-year survival. The cyan and red lines: the predictive accuracy determined by plotting the mean Kaplan-Meier estimates against the mean nomogram-predicted survival for individuals. Dots: the average predicted survival probability (*x* axis) and actual observed survival probability (*y* axis). Crosses: the bootstrap-corrected estimates. Vertical bars: 95% confidence intervals.

**Figure 7 biomedicines-13-01237-f007:**
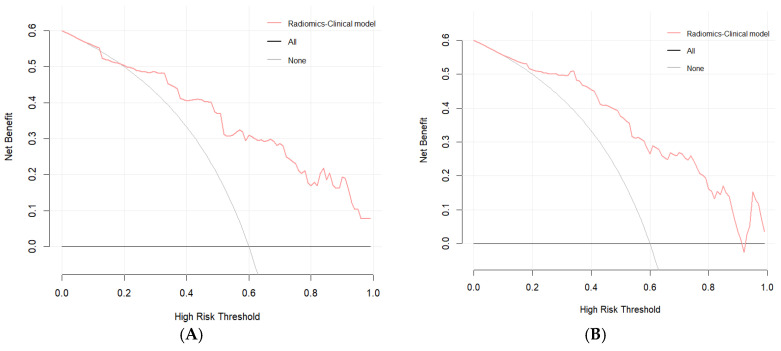
DCA plots for the radiomic–clinical model in predicting 1-year OS (**A**) and 3-year OS (**B**) in the training cohort, 1-year OS (**C**) and 3-year OS (**D**) in the internal validation cohort, and 1-year OS (**E**) and 3-year OS (**F**) in the external validation cohort.

**Figure 8 biomedicines-13-01237-f008:**
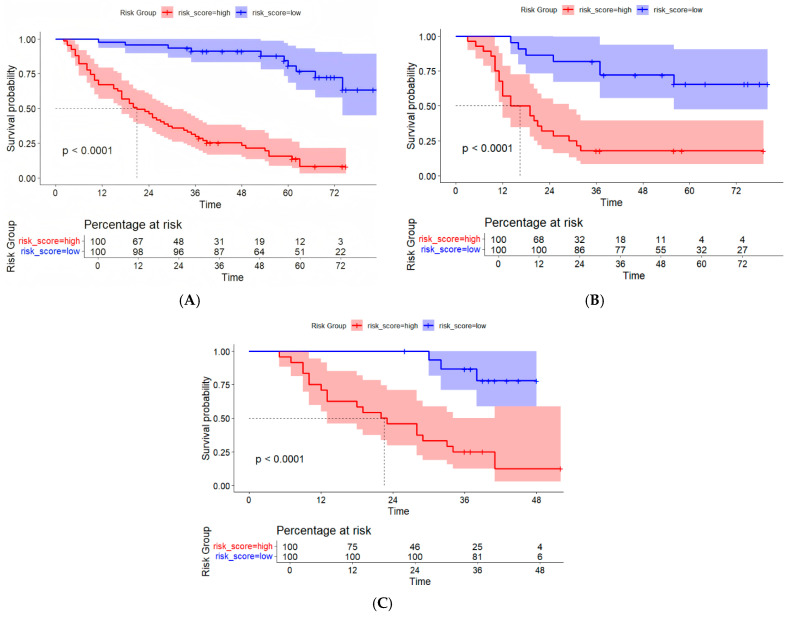
K–M curves of the radiomic–clinical model for predicting OS after HCC surgery. (**A**) Training cohort. (**B**) Internal validation cohort. (**C**) External validation cohort.

**Figure 9 biomedicines-13-01237-f009:**
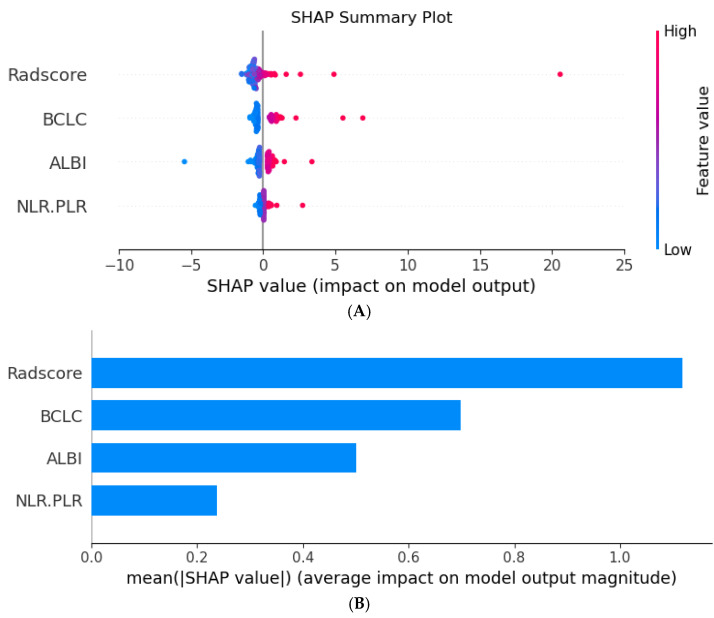
SHAP analysis plot of each predictor in the radiomic–clinical model. SHAP summary plot (**A**) shows the distribution of SHAP values of each predictor in all samples. The SHAP bar plot (**B**) shows the importance of each predictor in predicting the OS of patients after hepatectomy. The SHAP waterfall plot shows the predicted values generated for individual samples with the predicted outcomes as survival (**C**) and death (**D**).

**Table 1 biomedicines-13-01237-t001:** Clinical information of the patients in each cohort.

Variable	Training Cohort (N = 112)	Internal Validation Cohort (N = 50)	External Validation Cohort (N = 40)	*p*
Sex				0.242
Male	99 (88.4)	42 (84.0)	31 (77.5)	
Women	13 (11.6)	8 (16.0)	9 (22.5)	
Age (years)	56.8 ± 11.0	58.9 ± 8.9	56.45 ± 11.1	0.428
Maximum diameter of tumor (cm)	5.9 ± 2.8	6.8 ± 2.9	5.47 ± 2.1	0.054
Portal vein tumor thrombus				0.186
None	90 (80.4)	36 (72.0)	35 (87.5)	
Yes	22 (19.6)	14 (28.0)	5 (12.5)	
Satellite lesion				0.452
None	94 (83.9)	45 (90.0)	36 (90.0)	
Yes	18 (16.1)	5 (10.0)	4 (10.0)	
Cirrhosis of the liver				0.193
None	29 (25.9)	19 (38.0)	15 (37.5)	
Yes	83 (74.1)	31 (62.0)	25 (62.5)	
HBsAg				0.346
Negative	38 (34.0)	21 (42.0)	11 (27.5)	
Positive	74 (66.0)	29 (58.0)	29 (72.5)	
AFP (ng/mL)				0.350
≤400	80 (71.4)	31 (62.0)	30 (75.0)	
>400	32 (28.6)	19 (38.0)	10 (25.0)	
Child–Pugh				0.928
A	99 (88.4)	45 (90.0)	35 (87.5)	
B	13 (11.6)	5 (10.0)	5 (12.5)	
BCLCs				0.147
0 + A	78 (69.6)	32 (64.0)	33 (82.5)	
B + C	34 (30.4)	18 (36.0)	7 (17.5)	
NLR	2.6 (2.0, 3.8)	2.9 (2.2, 4.2)	2.4 (1.8, 3.3)	0.191
PLR	104.1 (78.3, 148.9)	107.6 (84.9, 141.9)	89.12 (64.5, 133.2)	0.290
NLR-PLR				0.275
0	52 (46.4)	20 (40.0)	25 (62.5)	
1	36 (32.2)	19 (38.0)	8 (20)	
2	24 (21.4)	11 (22.0)	7 (17.5)	
ALBI				0.216
Grade 1–2	60 (53.6)	34 (68.0)	22 (55.0)	
Grade 3	52 (46.4)	16 (32.0)	18 (45.0)	
AST (U/L)				0.497
≤40	51 (45.5)	20 (40.0)	21 (52.5)	
>40	61 (54.5)	30 (60.0)	19 (47.5)	
ALT (U/L)				0.833
≤50	80 (71.4)	38 (76.0)	29 (72.5)	
>50	32 (28.6)	12 (24.0)	11 (27.5)	
PT(s)				0.154
≤14	78 (69.6)	38 (76.0)	34 (85.0)	
>14	34 (30.4)	12 (24.0)	6 (15.0)	
Status				0.938
Survive	44 (39.3)	20 (40.0)	17 (42.5)	
Death	68 (60.7)	30 (60.0)	23 (57.5)	

**Table 2 biomedicines-13-01237-t002:** Univariate and multivariate Cox hazard regression analyses of clinical variables.

Variable	Univariate Cox		Multivariate Cox	
	HR (95%CI)	*p*	HR (95%CI)	*p*
Sex	1.14 (0.52–2.48)	0.750		
Age	0.99 (0.97–1.01)	0.404		
Maximum tumor diameter	1.12 (1.05–1.24)	0.002		
Portal vein tumor thrombus (PVTT)	3.55 (2.08–6.04)	<0.001		
Satellite lesion	2.26 (1.26–4.04)	0.006		
Cirrhosis of the liver	2.13 (1.16–3.92)	0.015		
HBsAg	0.94 (0.57–1.55)	0.807		
AFP	1.81 (1.09–3.00)	0.022		
Child–Pugh	3.02 (1.56–5.84)	0.001		
BCLC	4.00 (2.44–6.55)	<0.001	2.73 (1.62–4.62)	<0.001
NLR	1.13 (1.06–1.20)	<0.001		
PLR	1.01 (1.01–1.01)	<0.001		
NLR-PLR		<0.001		0.006
NLR-PLR (1)	2.45 (1.37–4.37)	0002	2.12 (1.17–3.85)	0.013
NLR-PLR (2)	4.39 (2.38–8.09)	<0.001	2.74 (1.42–5.26)	0.003
ALBI	2.19 (1.35–3.56)	0.001	1.94 (1.19–3.17)	0.008
AST	1.84 (1.14–2.96)	0.012		
ALT	2.26 (0.91–5.64)	0.080		
PT	1.86 (1.14–2.96)	0.014		

**Table 3 biomedicines-13-01237-t003:** Optimal radiomic features after screening.

Cohort	Image Type	Feature Class	Feature Name
NCE	Original	Glrlm	LongRunEmphasis
	Wavelet-HLH	Glrlm	GrayLevelNonUniformityNormalized
	Wavelet-HHL	Glszm	SmallAreaEmphasis
	Wavelet-HHL	Glszm	SmallAreaHighGrayLevelEmphasis
	Wavelet-HHH	Glcm	ClusterProminence
	Wavelet-LLL	Firstorder	Maximum
AP	Log-sigma-1–5-mm-3D	Glcm	DifferenceVariance
	Wavelet-LHH	Glcm	JointAverage
	Wavelet-HLL	Glszm	LowGrayLevelZoneEmphasis
	Wavelet-HLH	Gldm	DependenceEntropy
	Wavelet-HLH	Gldm	DependenceVariance
	wavelet-HHL	Glrlm	ShortRunLowGrayLevelEmphasis
PVP	Original	Shape	Sphericity
	Wavelet-LHL	Glszm	SmallAreaEmphasis

**Table 4 biomedicines-13-01237-t004:** Performance of each model in predicting OS after hepatectomy in the training cohort, internal validation cohort, and external validation cohort.

Model	Cohort	C-Index (95% CI)	1-Year AUC (95% CI)	3-Year AUC (95% CI)
Clinical model	Training	0.754 (0.696–0.812)	0.803 (0.705–0.901)	0.806 (0.725–0.888)
	Internal validation	0.695 (0.613–0.779)	0.794 (0.669–0.919)	0.826 (0.710–0.943)
	External validation	0.742 (0.640–0.844)	0.755 (0.551–0.959)	0.778 (0.619–0.937)
Radiomic model	Training	0.734 (0.672–0.795)	0.778 (0.672–0.885)	0.799 (0.718–0.880)
	Internal validation	0.715 (0.621–0.809)	0.763 (0.606–0.921)	0.855 (0.744–0.967)
	External validation	0.724 (0.627–0.820)	0.727 (0.569–0.886)	0.834 (0.698–0.971)
Radiomic–clinical model	Training	0.789 (0.734–0.845)	0.837 (0.741–0.932)	0.845 (0.773–0.917)
	Internal validation	0.726 (0.643–0.809)	0.801 (0.674–0.929)	0.880 (0.784–0.976)
	External validation	0.764 (0.663–0.865)	0.773 (0.609–0.937)	0.840 (0.700–0.980)

## Data Availability

The original contributions presented in this study are included in the article. Further inquiries can be directed to the corresponding author.

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
