# Peer review of "A CECT-Based Radiomics Nomogram Predicts the Overall Survival of Patients with Hepatocellular Carcinoma After Surgical Resection"

_biomedicines, 2025, doi:10.3390/biomedicines13051237_

Round 1
Reviewer 1 Report
Comments and Suggestions for Authors
In this paper, the authors investigate the value of a nomogram based on preoperative enhanced computed tomography (CT) radiomics features combined with clinical variables in predicting OS after HCC resection. Additionally, SHAP interpretability analysis was employed to further quantify the degree of contribution of each risk factor in the optimal model. The authors have conducted an interesting and relevant study. The following comments are provided for consideration, and the authors are encouraged to address and revise accordingly.
- How did the authors utilize DICOM format images during preprocessing prior to radiomics feature extraction?
- All CT images were imported into 3D Slicer for manual segmentation of tumor lesions. Why did the authors not use an automatic segmentation algorithm?
- How were the radiomics features extracted? Please clarify the extraction process for tumor morphological features, first-order statistical features, second-order texture features, and features derived from filter-based transformations.
- The authors should provide a more detailed explanation of Figure 2, particularly the steps outlined in the radiomics workflow diagram.
- A total of 1,130 features were extracted from the NCE, AP, and PVP phase images. Following dimensionality reduction using consistency analysis, Spearman correlation analysis, and LASSO Cox regression, 14 features with non-zero coefficients were retained. More details are needed regarding each step in this feature selection process.
- Given that this is a dual-center study with a relatively small sample size, how do the authors justify the generalizability and reliability of the proposed model?
- What experimental evidence supports the claim that "this model can serve as a valuable tool for clinicians in developing individualized treatment strategies for patients with HCC"? Please elaborate.
Author Response
Comments and Suggestions for Authors
In this paper, the authors investigate the value of a nomogram based on preoperative enhanced computed tomography (CT) radiomics features combined with clinical variables in predicting OS after HCC resection. Additionally, SHAP interpretability analysis was employed to further quantify the degree of contribution of each risk factor in the optimal model. The authors have conducted an interesting and relevant study. The following comments are provided for consideration, and the authors are encouraged to address and revise accordingly.
Point-by-point response
Re: Dear Editors and Reviewers: Thank you for reviewing our manuscript, which we are thankful for the opportunity to revise. On the basis of your highly constructive comments and suggestions, we have addressed all of your comments and suggestions. We have revised the paper and provided point-to-point responses to the issues raised in the peer review report. Please find our detailed responses below and the corresponding revisions shown in track changes in the resubmitted files.
1.How did the authors utilize DICOM format images during preprocessing prior to radiomics feature extraction?
Re: We checked and revised the Materials and Methods section of this manuscript. Please check.
All CT images were retrieved in DICOM format from the Picture Archiving and Communication Systems (PACS). To minimize the impact of different CT models on the experimental results, the original images were preprocessed before extracting radiomics features. This preprocessing included resampling (voxel size: 1 mm×1 mm×1 mm), image gray-level discretization (binwidth: 25) and image gray-level normalization to control image noise and standardize the voxel intensity.
2.All CT images were imported into 3D Slicer for manual segmentation of tumor lesions. Why did the authors not use an automatic segmentation algorithm?
Re: Three methods of segmentation are currently used in radiomic analysis to delineate target lesions: manual, semiautomatic and automatic segmentation. Among them, manual segmentation is the most common method. In this study, manual segmentation was used in accordance with the protocols in a previous study.
We have added relevant content for explanation in the discussion section.
3.How were the radiomics features extracted? Please clarify the extraction process for tumor morphological features, first-order statistical features, second-order texture features, and features derived from filter-based transformations.
Re: We revised the the Materials and Methods section. Please check.
Finally, 1130 radiomics features were extracted from the NCE, AP and PVP images, re
spectively. Which included seven classes: tumor morphological features (14 features),
first-order statistical features (216 features), gray level co-occurrence matrix (GLCM, 288 features), gray level run length matrix (GLRLM, 192 features), gray level size zone matrix (GLSZM, 192 features), neighborhood gray tone difference matrix (NGTDM,60 features), and gray level dependence matrix (GLDM,168 features).
4.The authors should provide a more detailed explanation of Figure 2, particularly the steps outlined in the radiomics workflow diagram.
Re: We provided a detailed explanation of Figure 2. Please check.
Radiomics workflow diagram.The CT images of all patients were imported into 3D-Slicer software in DICOM format for tumor segmentation. Then, radiomics features were extracted from the NCE, AP and PVP images, respectively. Following dimensionality reduction using consistency analysis, Spearman correlation analysis, and LASSO Cox regression, optimal features with non-zero coefficients were retained. Cox hazard regression analysis was used to construct models, and the model with the best performance was visualized as a nomogram.
- A total of 1,130 features were extracted from the NCE, AP, and PVP phase images. Following dimensionality reduction using consistency analysis, Spearman correlation analysis, and LASSO Cox regression, 14 features with non-zero coefficients were retained. More details are needed regarding each step in this feature selection process.
Re: Dear Reviewer, thank you for reviewing our manuscript and your highly constructive comments and suggestions. We are grateful for the opportunity to revise this manuscript.
We checked and added details. Please check the Materials and Methods section.
The intraobserver and interobserver consistencies of the features were evaluated by determining the intraclass correlation coefficients (ICCs). To calculate the intraobserver ICC, the CT images of 64 patients were randomly selected and segmented twice by radiologist A (PZ) within one month. To calculate the interobserver ICC, the selected CT images were independently segmented by two radiologists (radiologist A and radiologist B (YS)).Features with ICCs <0.75 were excluded. The feature with ICC>0.75 was considered to have good stability and consistency, and this feature was retained and entered the next step of screening.
The obtained radiomic feature data were processed via the Z score statistical method to eliminate the dimensionality differences of the data. Redundant features were removed by means of the Spearman correlation test. In this step, feature pairs with an absolute Spearman correlation coefficient above 0.8 were removed. The LASSO Cox regression algorithm was used to select the best features for predicting prognosis, and the Radscore was calculated according to the coefficient corresponding to each feature.
- Given that this is a dual-center study with a relatively small sample size, how do the authors justify the generalizability and reliability of the proposed model?
Re:We have checked and revised. Please check the Limitations section.
It is necessary to integrate data from multiple sources with larger sample sizes for further validation.
- What experimental evidence supports the claim that "this model can serve as a valuable tool for clinicians in developing individualized treatment strategies for patients with HCC"? Please elaborate.
Re:We are very sorry for the inappropriate wording of this sentence.We have checked and modified. Please check the Abstract section.
This model is beneficial for clinicians in developing individualized treatment strategies for patients with HCC.

Reviewer 2 Report
Comments and Suggestions for Authors
Technically the work has many advantages like
-
Comprehensive Model Integration:
-
Combines radiomics features from CECT images with clinical indicators (BCLC stage, ALBI, and NLR-PLR score), enhancing prediction accuracy.
-
-
Robust Model Performance:
-
The Radiomics-Clinical fusion model achieved high C-index and AUC values across training (C-index = 0.789), internal (0.726), and external validation cohorts (0.764), indicating good generalizability.
-
-
Use of SHAP Analysis:
-
Employs SHAP (Shapley Additive Explanations) to interpret the model, improving transparency and clinical trust.
-
-
Multiple Validation Cohorts:
-
Includes both internal and independent external validation, strengthening the model's reliability.
-
-
Radiomics Feature Optimization:
-
Feature selection through ICC filtering, Spearman correlation, and LASSO regression ensured dimensionality reduction and model efficiency.
-
-
Dynamic Nomogram Interface:
-
Developed a web-based dynamic tool for clinicians to estimate patient survival probabilities (Shiny app), facilitating clinical application.
-
-
Clinical Relevance:
-
Independent predictors used in the model (e.g., ALBI, BCLC, NLR-PLR) are already well-recognized in HCC prognosis.
-
Still, from a validation point of view, if the authors can improve on the following points, it will be advantageous.
Important prognostic factors like microvascular invasion (MVI), tumor differentiation, and genomic data were not included. If the author can include these parameters, it will improve the result.
-
Single-Modality Imaging:
-
Relied only on CT imaging; combining MRI or other modalities could potentially improve prediction.
-
-
Exclusion of Treatment Variability Post-Surgery:
-
No detailed analysis of post-surgical treatments (like TACE, sorafenib) which could affect survival outcomes.
-
Author Response
1,Still, from a validation point of view, if the authors can improve on the following points, it will be advantageous.Important prognostic factors like microvascular invasion (MVI), tumor differentiation, and genomic data were not included. If the author can include these parameters, it will improve the result.
Re: Dear Editors and Reviewers: Thank you for reviewing our manuscript, which we are thankful for the opportunity to revise. On the basis of your highly constructive comments and suggestions, we have addressed all of your comments and suggestions. We have revised the paper and provided point-to-point responses to the issues raised in the peer review report. Please find our detailed responses below and the corresponding revisions shown in track changes in the resubmitted files.
Because this was a retrospective study spanning a long period of time, pathological factors such as MVI, degree of tumor differentiation, and gene expression were not included. Those data should be included for further verification to improve the result.
Due to time deadline and other reasons, we will include the above parameters for further verification.We revised this paper in limitations section. Please check it.
Second, because this was a retrospective study spanning a long period of time, pathological factors such as MVI, degree of tumor differentiation, and gene expression were not included. Those data should be included for further verification to improve the result.
2,Single-Modality Imaging:
Relied only on CT imaging; combining MRI or other modalities could potentially improve prediction.
Re: Dear Reviewer, thank you for reviewing our manuscript and your highly constructive comments and suggestions. We are grateful for the opportunity to revise this manuscript.
Due to time deadline and other reasons, we will combine MRI or other modalities for further verification.
We revised this paper. Please check the Limitations section.
In addition, this study relied only on CT single-modality imaging, combining MRI or other modalities may improve prediction.
3,Exclusion of Treatment Variability Post-Surgery:
No detailed analysis of post-surgical treatments (like TACE, sorafenib) which could affect survival outcomes.
Re: Dear Reviewer, thank you for reviewing our manuscript and your highly constructive comments and suggestions. In this study, there is no detailed analysis of post-surgical treatments data which could affect survival outcomes. However, due to time deadline and other reasons, we will include postoperative data for further verification.We revised this paper. Please check the Limitations section.
Finally, there is no detailed analysis of postoperative data (like TACE, targeted therapy and immunotherapy) which could affect survival outcomes in this study.These needs to be included in future study.
